# Microwave Characterization and Modelling of PA6/GNPs Composites

Erika Pittella [1,*], Emanuele Piuzzi [1], Pietro Russo [2] and Francesco Fabbrocino [3]

1   Department of Information Engineering, Electronics and Telecommunications, Sapienza University of Rome, Via Eudossiana, 18, 00184 Roma, Italy; emanuele.piuzzi@uniroma1.it
2   Institute for Polymers, Composites and Biomaterials (IPCB), National Research Council, Via Campi Flegrei, 34, 80078 Pozzuoli, Italy; pietro.russo@unina.it
3   Department of Engineering, Telematic University Pegaso, Piazza Trieste e Trento, 48, 80132 Naples, Italy; francesco.fabbrocino@unipegaso.it
*   Correspondence: erika.pittella@uniroma1.it

**Abstract:** The interest in composite materials has increased in the last decades since they have the advantages of combining intrinsic properties of each component and offer better performance with respect to the base constituents. In particular, these kinds of materials can have different electrical characteristics by varying the filling percentage and, therefore, they can be used in diverse applications. Thus, a detailed study of the microwave response of these composite systems is of great practical importance. In fact, the dielectric constant and loss tangent are key factors in the design of microwave components. In this frame, the outstanding properties of graphene-like fillers may be exploited to develop new very interesting materials to study and characterize. In this paper, microwave characterization of compounds, based on nylon 6 containing different percentages of graphene nanoplatelets, is carried out taking the neat matrix sample processed under the same conditions as benchmark. The measurements were carried out using two microwave systems, operating at two different frequency bands, appropriate to characterize solid and compact material samples. The achieved results, in line with limited data from the literature and from material data sheets, highlight the possibility to use the present polymers as an excellent electromagnetic interference shielding, as confirmed by full wave electromagnetic numerical simulations that were conducted with a numerical electromagnetic software.

**Keywords:** graphene nanoplatelets; polyamide 6; composites; microwave characterization; complex permittivity; modeling





## 1. Introduction

Graphene is a material consisting of single layers of carbon about 0.34 nm thick organized in a two-dimensional honeycomb lattice [1,2]. Commonly multiple layers of graphene are superimposed to form agglomerates due to van der Waals interfacial interactions and better known as graphene-like structures. This behavior strongly influences their dispersion when used as fillers to enhance the functionality of various host matrices.

In the light of preliminary studies that have clearly highlighted the extraordinary properties of individual graphene layers mainly in terms of lightness [3], high thermal and electrical conductivity [4], flexibility [5], mechanical strength [6,7] and optical properties [8], an enormous interest has been directed towards these materials not only by academic and industrial research but also at the legislative level. With reference to this last aspect, it is worth mentioning the billionaire loan from the European Community to a major project better known as Graphene Flagship aimed at extending the production and use of graphene on an industrial scale.

Potential prospects for use of graphene-like materials concern, among others, flexible and transparent electronic devices (EMI shielding [9], antennas [10], supercapacitors [11])

wearable electronics [12], smart sensors [13], efficient solar panels, targeted drag-delivery, healthcare and construction. Additional recent papers are related to the applications of graphene composites in different engineering application fields [14–16]. In [14], some new concepts for graphene/SiC composites are suggested; in particular, the mechanical and electrical properties of these composites are theoretically analyzed by the two-fractal theory and experimentally verified by open experimental data. The optimization of SiC/graphene composites is addressed in [15] with the aim of maximizing specific material properties under the constraint of a given strength. In [16], a simple and effective analytical approach of nonlinear buckling of functionally graded graphene reinforced composites (i.e., advanced composite materials in which some characteristic properties are continuously varied through the thickness of the structure) is presented.

Moreover, nowadays, the interest in developing microwave absorbing materials based on polymers with light weight, high specific strength, stable properties and outstanding mechanical properties is increasing enormously [17]. Polymer based EMI shielding materials are mainly obtained by the template method [18], the freeze-drying method [19], the hydrothermal method [20] and the foaming method [21].

More specifically, in template methods the conductive networks are constructed on open-celled foams with 3D cellular structures: for example, in [18] carbon foam was prepared by pyrolysis of lignin–resorcinol–glyoxal precursor, using flexible polyurethane foam as template.

The freezing-drying method considers the conductive fillers in suspension usually assembled and frozen to form ice crystals or solid organic molecules; as an example, in [19] a CNTs/polyimide foam is prepared by freeze-drying the mixture of CNTs with polyamic acid followed by thermal imide.

In hydrothermal methods, instead, the reaction system is heated and pressurized to improve the activity of reactants and promote the self-assembly or lapping between the conductive fillers; highly conductive and superelastic graphene/CNTs hybrid foams were assembled by this method in [20].

Finally, the foaming method mainly introduces a large amount of gas into the composites to form the mutually penetrated or closed cellular structures. A lightweight biodegradable poly (l-lactic acid) (PLLA)-multiwalled carbon nanotubes nanocomposite foam using a combinatorial technology of pressure-induced flow processing and supercritical carbon dioxide (Sc–$CO_2$) foaming is presented in [21].

The interest in graphene-based EMI shielding for high performance electromagnetic wave attenuation, to be used in commercial, military and electronic devices, is growing [22], especially in the GHz frequency band [23]. Details on the fabrication of graphene-based EMI shielding are included in [24,25]. In [26] the first experimental results on EMI shielding effectiveness (SE) [27] of monolayer graphene was conducted showing a value of 2.27 dB. Few-layer graphene (FLG), obtained from direct exfoliation of graphite, was fabricated into paraffin wax to prepare FLG/wax composites and investigate their EMI shielding performance [28] finding a similar SE equal to 5.5 dB; in [29] reduced graphene oxide (RGO) sheets interleaved between polyetherimide (PEI) films fabricated by electrophoretic deposition (EPD) are studied: incorporating only 0.66 vol % of RGO, composite films exhibited an EMI SE of 6.37 dB.

Despite the considerable efforts already made, challenges remain to be addressed and further potentials still need to be adequately explored. Actually, although many methods have been developed over time to produce graphene-like materials, production volumes are still limited and this aspect, by holding back the reduction in price, limits their range of large-scale applications [30].

In this frame, Graphene NanoPlatelets (GNPs) combine large scales production and low costs with remarkable physical properties [31]. GNPs are made of single and few graphene layers mixed with thicker graphite. Therefore, GNPs thickness can reach up to 100 nm within the same production batch.

Regarding the identification of further fields of application of graphene-based materials, at present, for example, only a few studies are available regarding the effect of the content of GNPs on the microwave dielectric properties of filled polymers. Therefore, with the main purpose of bridging this gap, in this paper a microwave characterization of compounds based on polyamide 6 (PA6) and containing 1%, 3% and 5% by weight of GNPs is presented.

Moreover, electromagnetic simulations were conducted with Microwave Studio from Computer Simulation Technology (CST) [32] to evaluate this kind of material as an EMI shield, simulating inside the CST CAD a material with the measured characteristics of the PA6-GNPs composite.

The paper is organized as follows: first the materials and methods used for the measurements are presented and then the experimental set-up is discussed. Subsequently, the results are shown and properly discussed, also proceeding with CST simulations, before drawing some conclusions.

## 2. Materials and Methods

The considered samples are constituted by polyamide PA6 and graphene nanoplatelets. PA6 is an industrial grade resin, offering an exceptional balance between excellent mechanical properties and low price.

In this study, the polyamide matrix was provided by Ravago [33] under the trade name Ravamid B NC (density: 1.09 g/cm$^3$—HDT: 50 °C) while GNPs (density: 2.2 mg/cm$^3$—Young's modulus: 1 TPa—Tensile strength: 5 GPa) were furnished by Nanesa S.R.L. in the form of flakes (diameter: 10 ÷ 20 μm—thickness: 5 ÷ 20 nm) [34].

Samples containing 1%, 3%, 5% by weight of GNPs were obtained by melt-compounding with the aid of a co-rotating twin-screw extruder, Collin Teachline ZK25T, and transformed in plaques by using a lab Press Collin model P400E.

Neat matrix samples, prepared with the same procedure, were considered as reference.

The specimens to be tested were cut from the press-moulded plates in appropriate shape and size, imposed by the chosen measurement set-up, constituted by two rectangular waveguides. In particular, the specimens had to be inserted in the transversal section of two parallelepiped waveguide sample holders. The dimensions of the waveguide sample holders are established by the transversal section of the used rectangular waveguide system, the WR430 and WR90 (working frequency band: 1.7–2.6 GHz, 8.2−12.4 GHz, respectively):

(1)　109.22 mm × 54.61 mm (WR430)
(2)　22.86 mm × 10.16 mm (WR90)

Parallelepiped samples were cut from these plaques with a thickness t = 2.2 mm (see Figure 1).

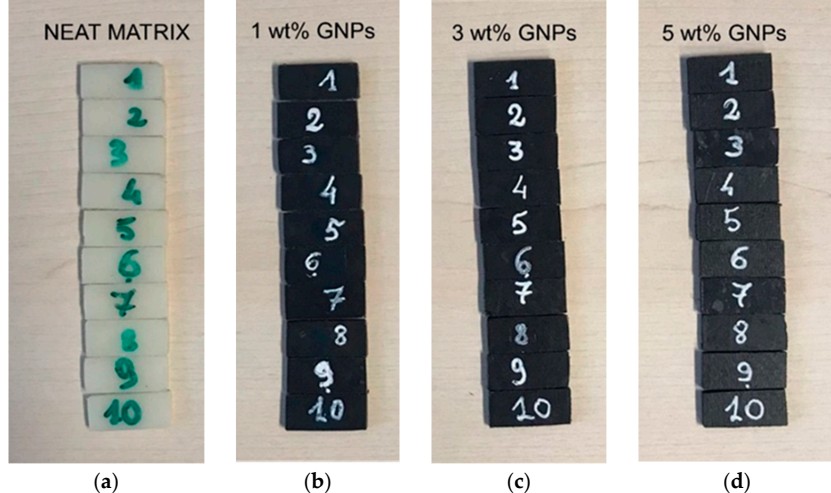

**Figure 1.** Neat matrix (**a**), GNPs 1% (**b**), 3% (**c**), 5% (**d**) by weight samples.

### 2.1. Methods

The dielectric properties of fabricated samples were measured using the transmission line permittivity measurement method, described in detail in [35], that allowed to obtain broadband measurement of the dielectric parameters. In this method, a sample is placed in a section of a waveguide or a coaxial line and the scattering parameters are measured by the vector network analyzer (VNA). The relevant scattering equations relate the measured scattering parameters to the permittivity and the permeability of the material. For the measurement model, the equation system contains as variables the complex permittivity and permeability, the two reference plane positions and, in some applications, the sample length. The standard NIST method presents a procedure to determine the complex permittivity from the scattering equations which is stable over the frequency spectrum, minimizing instabilities in determination of permittivity and allowing measurements for arbitrary length samples [35].

In our case, the transmission line is a rectangular waveguide. After inserting the specimen inside the sample holder an electromagnetic wave is sent at the waveguide input port. The complex permittivity of the specimen $\varepsilon_c = \varepsilon' - j\varepsilon''$, with $\varepsilon'$ real part of the permittivity that represents the ability of the material to store electric energy and it is related to the polarization, and $\varepsilon'' = \sigma/(\omega\varepsilon_0)$ imaginary part, representing the loss in the material and being responsible for the damping of the wave and dissipation of energy, is obtained by measuring the scattering parameters at the waveguide ports (see Figure 2) with a VNA, and then using a specific measurement model.

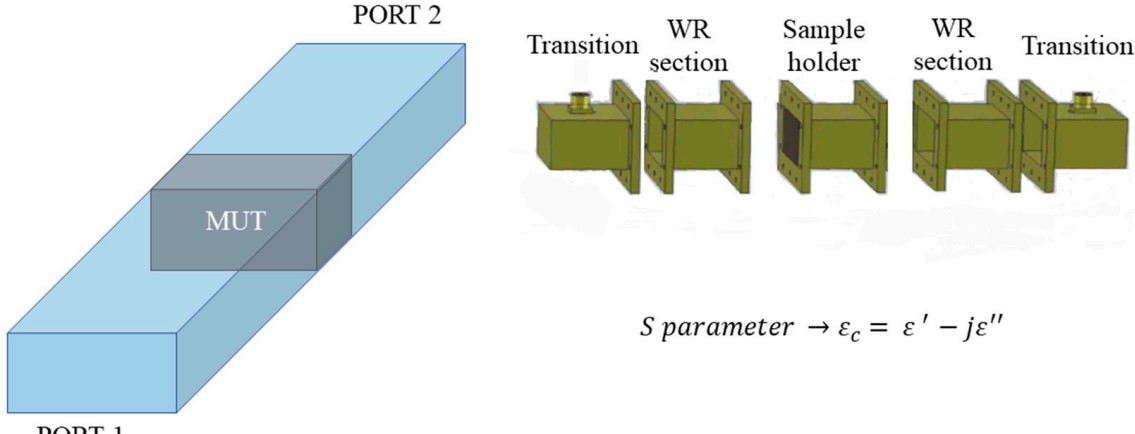

**Figure 2.** Scheme of the measurement method: after inserting the sample inside the sample holder an electromagnetic wave is sent at the waveguide input port and the complex permittivity of the sample $\varepsilon_c$ is obtained from the *S* parameters.

Measurements were performed using the vector network analyzer (VNA) Agilent E8363C, equipped with the Agilent 85071E software [36], which provided various measurement models, among which the Nicolson–Ross–Weir [37,38] and the National Institute of Standards and Technology (NIST) [39,40] models.

Specifically, results presented in this paper were obtained with the NIST model, since it is the most precise for nonmagnetic materials, such as those of our interest, and, moreover, it uses a mathematical formulation that limits errors associated with sample resonances.

### 2.2. Experimental Set-Up

Two waveguide systems were used to obtain the sample complex permittivity. One system covered the 1.7–2.6 GHz frequency band (WR430 waveguide) and the other one the 8.2−12.4 GHz (WR90 waveguide). This choice allowed to cover two microwave frequency bands. Moreover, these systems were suitable to characterize successfully solid and compact materials [41].

The WR430 waveguide system, composed of a couple of N-type coaxial to rectangular waveguide transitions, two standard waveguide sections long enough to guarantee that higher order modes were effectively attenuated, along with a sample holder, as shown in Figure 3, where also the VNA is visible.

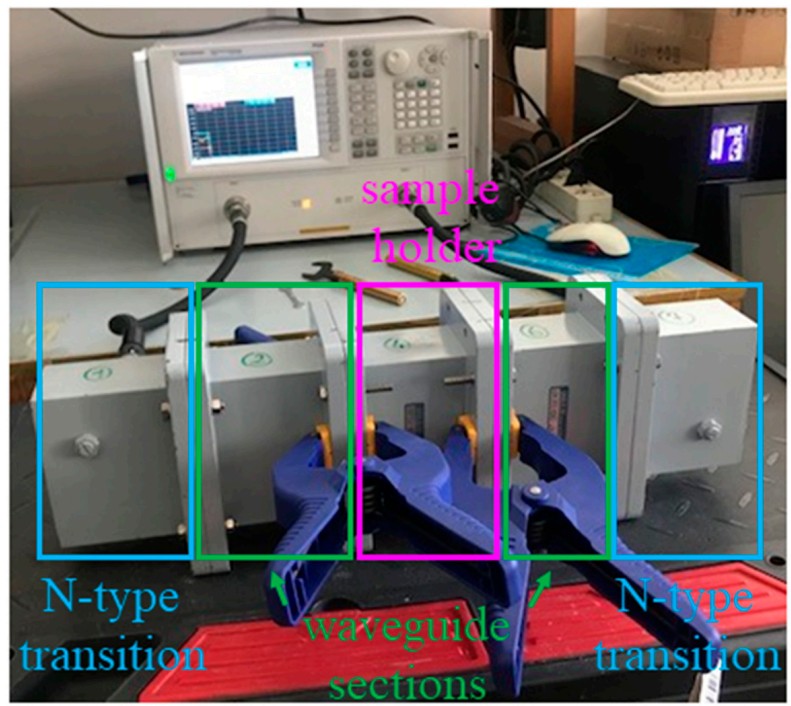

**Figure 3.** Measurement set-up with the WR430 system and the VNA.

The WR90 waveguide system was based on commercial components, while the sample holder was manufactured with a milling machine with 0.005 mm repeatability [42]. The WR90 measurement set-up is shown in Figure 4.

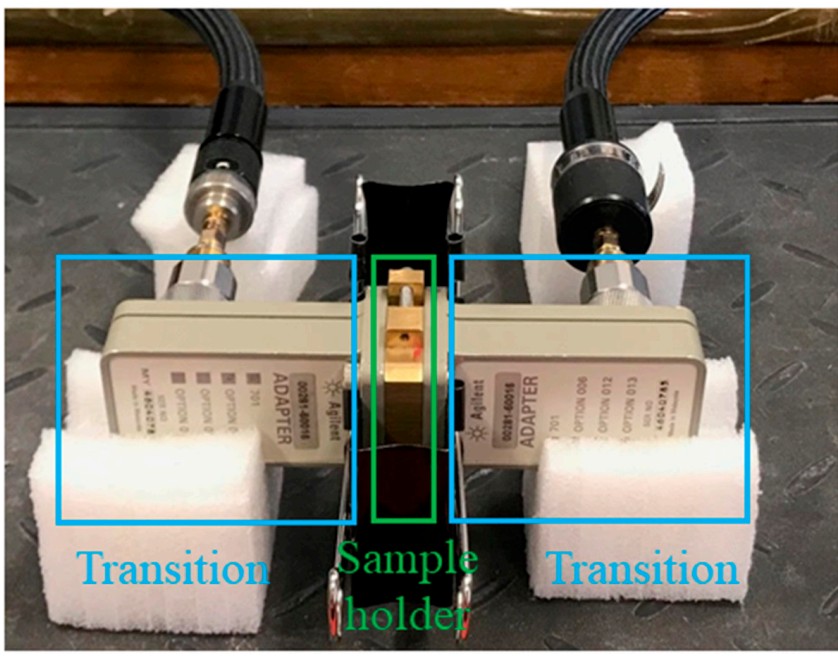

**Figure 4.** WR90 system measurement set-up.

In order to compensate for most systematic errors, the technique of vector error correction at the sample holder ports was applied, as described in [43], before performing S-parameter measurements. In particular, the vector error correction procedure characterized systematic errors, caused by imperfections in the test equipment and test setup (e.g., cables and connectors), by measuring known calibration standards, storing these measurements within the VNA memory and using these data to calculate an error model. This allowed us to use this error model to remove the systematic error effects, resulting in very accurate measurements.

## 3. Results

### 3.1. WR430

Measurements performed by inserting all the ten samples shown in Figure 1 in the WR430 waveguide system ( 1.7–2.5 GHz span ), one at a time, are reported in Figures 5 and 6 in terms of permittivity as a function of frequency averaged over the ten samples. In particular, Figure 5 shows the real part ($\varepsilon'$) and Figure 6 the imaginary part ($\varepsilon''$) of the complex permittivity for the PA6 neat matrix material and the other compounds explored—i.e., 1 wt%, 3 wt% and 5 wt% GNPs.

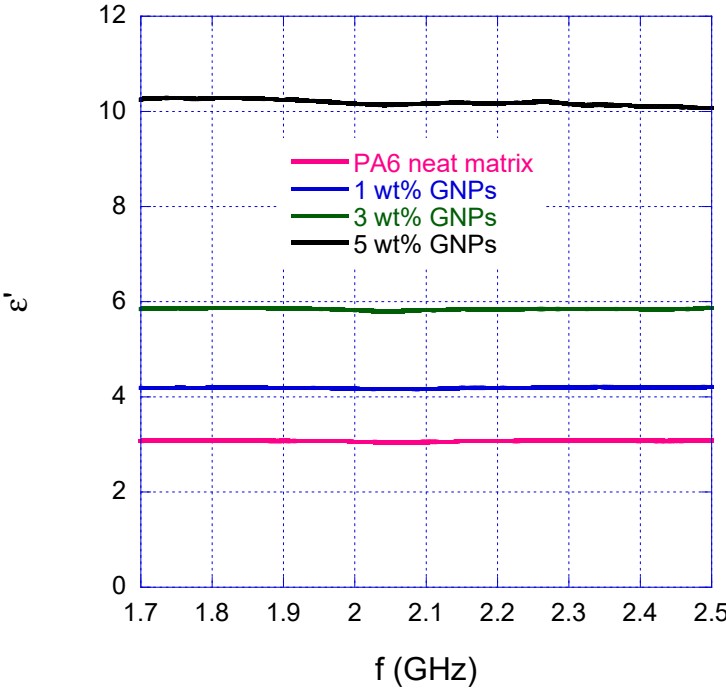

**Figure 5.** Averaged real part of the complex permittivity measured with the WR430 waveguide system.

The permittivity real part of the PA6 (neat matrix) has an average value over the explored frequency band of $\varepsilon'$ = 3.0723 with a standard deviation equal to $\sigma_\varepsilon$ = 0.0095. For PA6, no literature data corresponding to the frequency bands studied in this work are available. Indeed, the PA6 dielectric constant values can be found mainly at lower frequencies reported in material datasheets. To have an idea, at 1 kHz a value around 8 is declared [44], while in [45] a value of about 4–5 is stated at a frequency equal to 1 MHz. In [46], the authors measured the complex dielectric permittivity of an extensive variety of polymers, including the polyamide 6, at higher frequency, precisely at Q-band (30–50 GHz). The measured values are in line with our results, both in term of real part and imaginary part. In fact, as regards the PA6 averaged tangent loss $\left( \tan g\delta = \frac{\varepsilon''}{\varepsilon'} \right)$ value reported in the present paper, i.e., $3.5 \times 10^{-3}$ agrees with [39].

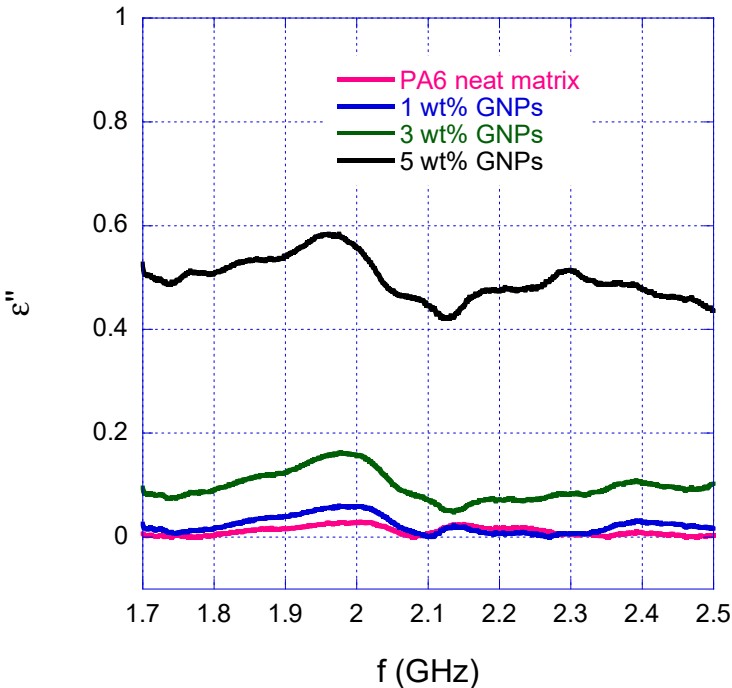

**Figure 6.** Averaged imaginary part of the complex permittivity measured with the WR430 waveguide system.

As for the compounds, the imaginary part $\varepsilon'' = \sigma/(\omega\varepsilon_0)$ increases with the GNPs percentage from a value of 0.0107 to 0.493 (i.e., 50 times higher in the 5 wt% sample case). This effect highlights the high efficiency of GNPs as fillers capable of enhancing the conductivity of polymer matrices, even if included in relatively modest quantities, thus increasing the ohmic losses due to the peculiarity of the material that causes energy dissipation.

It is worth noting here that the wavy nature of the curve in Figure 6 can be attributed to the non-perfect planarity of the sample probably related to the fabrication process and to the possible presence of small air-gaps within it. In fact, an inhomogeneous sample of high dielectric constant could cause the excitation of higher order modes giving rise to non-linear effects of the system [35].

For the compounds, it was not possible to directly compare the results obtained with the literature. In fact, to the best of the authors' knowledge, the few available works that include similar characterizations and analyze the variation of complex permittivity as a function of frequency, have essentially been focused on thermosetting (epoxy) matrix composites [47], usually able to guarantee better control of the dispersion of the filler than the thermoplastic counterparts. Similar studies are also available on materials consisting of carbon nanotubes added in polylactic acid (PLA), poly (butylene adipate-coterephthalate) (PBAT) and polyamide 66 by melt mixing [48,49]. Even the results, obtained in the latter cases at frequencies outside the range considered for this study, are not comparable with the ones discussed in this work.

### 3.2. WR90

Results found by using the WR90 waveguide system ($8.2 - 12.4$ GHz span) on reference neat matrix and PA6/GNPs samples are shown in Figures 7 and 8.

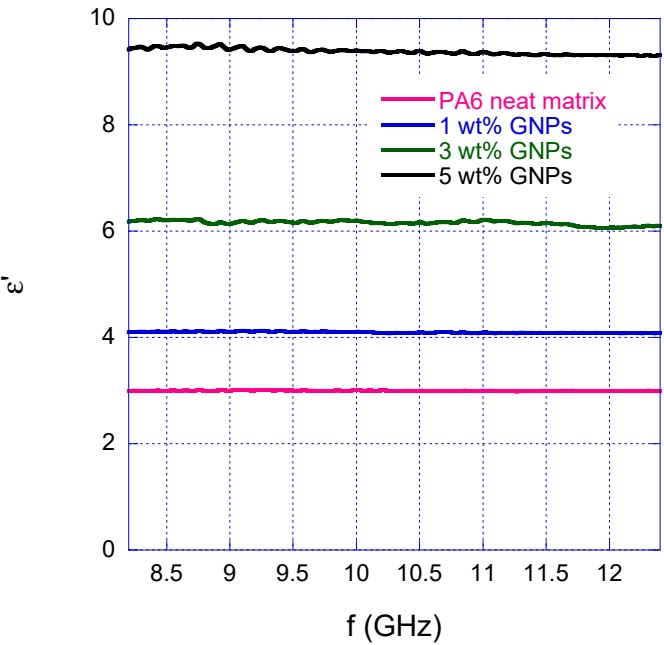

**Figure 7.** Averaged real part of the complex permittivity measured with the WR90 waveguide system.

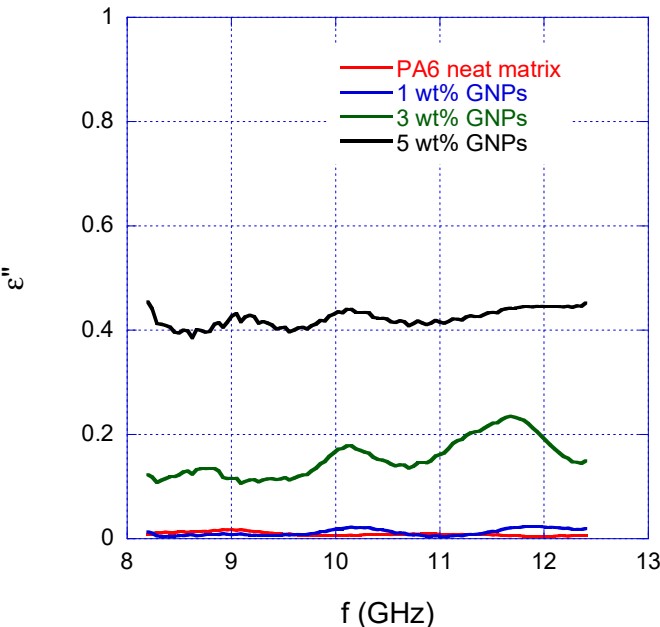

**Figure 8.** Averaged imaginary part of the complex permittivity measured with the WR90 waveguide system.

For the relative permittivity of the PA6 (Figure 7) an average value of $\varepsilon' = 2.999$ with a standard deviation equal to $\sigma_\varepsilon = 0.0059$ were found. These values can be compared with ones obtained in [50], but considering that in that case the value is measured slightly above the bandwidth considered in this paper. In particular, a value equal to 2.8 is reported for the real part; and this value corresponds to the higher working frequency range of 12.4–18 GHz, since measurements in [50] were performed with a WR62 waveguide system. Moreover, it is worth noting that the experiments on 3D-printed nylon 6 were conducted with different filling configurations and filling density, and the value equal to 2.8 corresponds to a 100% PA6 filling density, while lower values are reported (between 1.4 and 1.8), if a certain percentage of air is present in the considered samples.

As in the WR430 system, an increase of a factor 50 in $\varepsilon''$, and therefore in the conductivity $\sigma$, is found for the 5 wt% compound. However, even these data are not directly comparable with those already available in the literature being obtained in different frequency intervals.

In general, a slightly decreasing trend of $\varepsilon'$ and $\varepsilon''$, as a function of the frequency, can be noted.

The same considerations that were made for Figure 6 on the wavy nature of the curve can be deduced for Figure 8 also (i.e., non-perfect planarity of samples and possible presence of small airgaps).

### 3.3. Global Results

Complex permittivity overall results obtained using the two waveguide rectangular systems, together with the standard deviations are summarized in Table 1.

**Table 1.** The material averaged complex permittivity obtained by the two considered systems, together with the standard deviations.

|  | $\varepsilon'$ | | $\varepsilon''$ | |
|---|---|---|---|---|
|  | WR430 <br> 1.7–2.5 GHz | WR90 <br> 8.2–12.4 GHz | WR430 <br> 1.7–2.5 GHz | WR90 <br> 8.2–12.4 GHz |
| **PA6** | $3.0723 \pm 0.0095$ | $2.999 \pm 0.0059$ | $0.0107 \pm 0.0083$ | $0.0091 \pm 0.0045$ |
| **1%** | $4.195 \pm 0.019$ | $4.100 \pm 0.014$ | $0.0227 \pm 0.016$ | $0.0120 \pm 0.0083$ |
| **3%** | $5.846 \pm 0.038$ | $6.159 \pm 0.043$ | $0.106 \pm 0.042$ | $0.155 \pm 0.043$ |
| **5%** | $10.172 \pm 0.069$ | $9.381 \pm 0.060$ | $0.493 \pm 0.039$ | $0.422 \pm 0.030$ |

Looking at the table values, a very slight decreasing trend in frequency of both permittivity components can be observed for all the parameters (real and imaginary parts); however, with the exception of the 3 wt% case that has very close values in the two bands.

The achieved results appear to be interesting considering that a material with a high permittivity polarizes more in response to an applied electric field than a material with low permittivity, storing more energy.

Moreover, considering the neat matrix samples and the 5 wt% GNPs one it can be noted a significant increase in both the imaginary and complex part of the permittivity for both waveguide rectangular systems with increasing GNPs content, at least in the range of compositions studied so far. This suggests that since conductivity ($\sigma = \omega \varepsilon_0 \varepsilon''$) increases by the same quantity, the PA6/GNPs composite appears a good candidate as an EMI shielding material. In particular, the values of the conductivity increase from 0.001 S/m (neat matrix) to 0.06 S/m (5 wt% GNPs) in the WR430 frequency band, and from 0.005 S/m (neat matrix) to 0.3 S/m (5 wt% GNPs) in the WR 90 frequency band. Even if the value is obviously lower then conductivity value of metal shields, it could be sufficient, depending on the application, and at the same time it presents the innumerable advantages of composites (e.g., lightweight, structural efficiency and mechanical properties); moreover, the percentage of graphene nanoplatelets can be varied to increase, if necessary, the conductivity value of the considered composite). To better evaluate the shielding effectiveness of the studied material, full-wave EM simulations with CST Microwave Studio were conducted [32].

### 3.4. EM Simulations

Numerical analysis was carried out simulating an incident plane wave impinging on a parallelepiped made of a material with the same dielectric properties of the measured PA6-5 wt% GNPs composite (see CST 3D schematic in Figure 9). For the material, the dispersion fit is based on a general $n^{th}$ order model [51–53]. In particular, a list of $\varepsilon'$ and $\varepsilon''$ values can be defined by different frequency points, allowing an efficient and accurate material model.

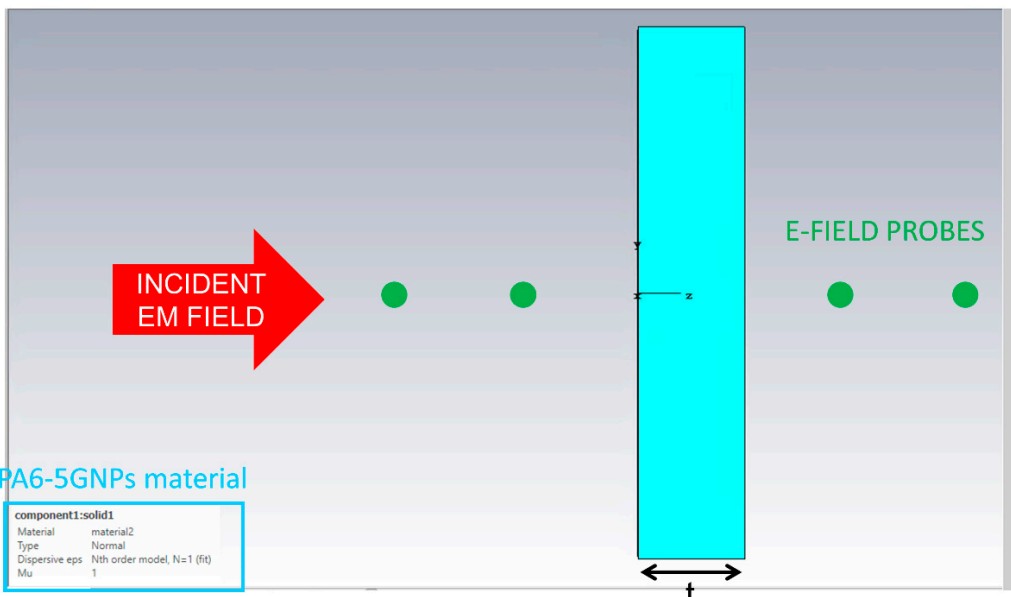

**Figure 9.** CST 3D schematic: a parallelepiped made of PA6-5GNPs composite is designed and electric field probes and an incident electromagnetic field impinging on the material are considered in the simulation.

The parameters of these general models can be directly defined in the software Dispersive Material Parameters dialog or by applying an automatic fitting scheme to a list of material data values in the Dielectric/Magnetic Dispersion Fit dialog of the CST software. Inserting the $\varepsilon'$ and $\varepsilon''$ values as a function of the frequency, the dispersive model of the material can be defined in the form of the first-order Debye model [54], that describes the relaxation response of a dielectric medium to an external oscillating electric field in terms of permittivity as a function of frequency. The model is characterized by the following formulation for the relative permittivity:

$$\varepsilon(\omega) = \varepsilon_\infty + \frac{\varepsilon_S - \varepsilon_\infty}{1 + j\omega\tau} \tag{1}$$

where the parameters of the model are: $\varepsilon_\infty = 9.2$, $\varepsilon_S = 10.4$ and $\tau = 4.02 \times 10^{-11}$ s.

Electric field probes are inserted in the CAD and an incident electromagnetic field was simulated impinging on the material (Figure 9).

The Transient Solver was used among the solvers available for the simulation. This time domain solver is very efficient for most high frequency applications. It is based on the Finite Integration Technique (FIT) and, in combination with the Perfect Boundary Approximation (PBA)® feature and the Thin Sheet Technique™ (TST) extension, is able to increase substantially the accuracy of the simulation, in comparison to other techniques employing a conventional hexahedral mesh.

The simulation structure has to be defined within a bounding box, therefore a boundary condition for each plane has to be specified. In particular, the open add space condition is set, that extends the geometry virtually to infinity, with some extra space added between structure and applied boundary condition, that is the perfectly matched layer (PML), with respect to the open boundary conditions. The number of mesh cells used in this simulation is approximately 53.

Results show that a brick of thickness t = 4 cm can attenuate the EM field of about 30 times around 9 GHz (Figure 10). This means that if an EM field impinges on a parallelepiped made of PA6-5GNPs composite, the SE = 20 log (Ei/Et) [26] is equal to 20 dB: a value that is comparable and, in some cases, higher than shielding effectiveness of compos-

ite materials reported in [55], where an overview of the SE of different composite materials is given, showing values between 3–28 dB [56], 25–50 dB [57] and 25–30 dB [58].

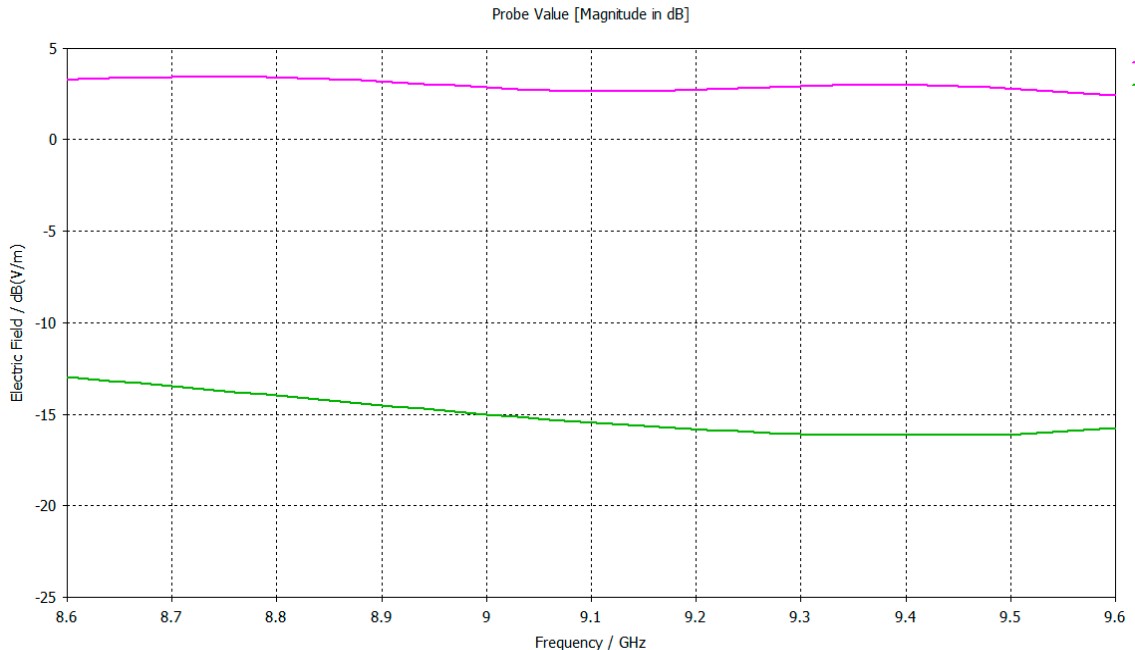

**Figure 10.** Electric field (dBV/m) computed at the electric field probes in front and behind the material parallelepiped.

Compared to commonly used shields, composite materials have a lower SE, but other advantages, such as a low weight, that must be considered when choosing the shielding material to use, depending on the application. Indeed, combining two or more materials leads to the possibility of obtaining an enhanced material in terms of electrical and mechanical properties, aligned with actual trends of economic and environmental global strategies. As an example, the aerospace industry is interested in this type of materials due to lightweight and structural efficiency, and other industry fields from automotive to medical equipment can benefit from these characteristics. In particular, mechanical performances of considered composites are discussed in [59]. In particular, Russo et al. showed that the inclusion of GNPs, on the range explored so far, generates a monotonous growth of the host matrix stiffness both in tensile (+13%) and, above all, in bending with average values increasing from 1196 MPa for neat polyamide to 3303 MPa (+276%) for the sample filled with 5 wt% -of GNPs.

Vice versa, for situations of quasi-static loading, traction and bending, a non-monotonous trend of the strength was detected with average values maximized for the sample containing 3% by weight of GNP and in any case, in bending, always higher than the value characterizing the neat matrix. Considering the already consolidated wide use of PA 6 in many industrial sectors, such as transport and electronics in which the employment of EMI shielding systems is increasingly required, the further improvement of the mechanical performance allows us to consider that the new formulations can certainly satisfy the requirements of the reference markets.

The above considerations suggest that the considered material can be used as an EM shield.

## 4. Conclusions

In this paper, a microwave characterization of composites based on PA6 and containing 1 wt%, 3 wt% and 5 wt% GNPs is carried out, with a measurement set-up constituted by WR430 and WR90 waveguide systems, and the VNA Agilent E8363C, equipped with a software implementing the NIST model for dielectric measurements. Moreover, numerical

EM simulations were conducted with MWS by CST to evaluate the properties of the material as an EMI shield.

The main results obtained so far can be summarized as follows:

(a) regardless of the waveguide system used (WR430, WR90), a slightly decreasing trend in frequency can be observed for all the parameters (real and imaginary parts) with values that are in line with the few results that can be found in literature;

(b) a 4 cm thick parallelepiped whose dielectric characteristics are equal to the ones measured for the 5 wt% samples, attenuates the E-field of 20 dB @9 GHz.

These considerations confirm that, thanks to the high electrical conductivity, a graphene plane yields a good shielding efficiency against microwave radiation.

As future developments, it would be very interesting to investigate these materials in broader band experiments. This can be performed, for example, with a broad band probe such as the Keysight HTP [60] that has a frequency range from 200 MHz to 20 GHz. The broad band study first could further validate our results, and secondly it could be used to find out an analytic model to characterize the polymers [61]. Furthermore, it will be interesting studying the possibility to use dielectric models for heterogeneous system, as for example the Maxwell model, in order to predict the permittivity of the compounds [62]. Finally, another important aspect that cannot be left out is the mechanical properties investigation of the considered samples.

**Author Contributions:** Conceptualization, E.P. (Erika Pittella) and P.R.; methodology, E.P. (Erika Pittella) and E.P. (Emanuele Piuzzi); software, E.P. (Erika Pittella) and E.P. (Emanuele Piuzzi); validation, E.P. (Erika Pittella) and F.F.; formal analysis, E.P. (Emanuele Piuzzi); investigation, E.P. (Erika Pittella) and E.P. (Emanuele Piuzzi); resources, E.P. (Erika Pittella) and F.F.; data curation, E.P. (Erika Pittella); writing—original draft preparation, E.P. (Erika Pittella); writing—review and editing, E.P. (Erika Pittella) and P.R.; visualization, E.P. (Emanuele Piuzzi); supervision, F.F.; project administration, F.F.; funding acquisition, E.P. (Erika Pittella). All authors have read and agreed to the published version of the manuscript.

**Funding:** This research was partially funded by National Operational Program for Research and Innovation 2014–2020—European Social Fund, Action I.2 "Attraction and International Mobility of Researchers", grant number AIM1857230-2.

**Conflicts of Interest:** The authors declare no conflict of interest.

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
