# Peer review of "Microwave Characterization and Modelling of PA6/GNPs Composites"

_mca, doi:10.3390/mca27030041_

Round 1
Reviewer 1 Report
While the reading of the abstract let expect sevarl innovative points, I must admit that I am disconcerted by the content of this work. I do not recommend publication of this manuscript because it does not add any substantial physical content to the existing literature. That is not applied physics at the level we expect. This manuscript appears to contain a presentation of data without the in-depth analysis expected. Hence it is not sufficient to report outcomes of experiments without drawing out new physics in the interpretation of the results. In this manuscript, there are descriptions about the Figures, but there is little said about their physical significance and what new understanding can be gained from them.
Many studies dealing with the absorption properties of composites containing graphene filler particles are not cited and eventually not discussed.
While Figs. 1-4 may be adapted for a technical report, they are irrelevant to be included in a paper of the archival literature.
Figs. 4-8 are just described and not discussed in physical terms/ What is (are) the physical mechanism(s) of loss? Unclear. What is the physical reason of the wavy nature of the curves in Figs. 6 and 8? Not discussed.
I do not get the point for the use of Fig. 9. How Eq. (1) is justified in physical terms? Unclear What value of N is chosen and argumented? How many constants (values?) are used in Eq.(1)?
Author Response
While the reading of the abstract let expect sevarl innovative points, I must admit that I am disconcerted by the content of this work. I do not recommend publication of this manuscript because it does not add any substantial physical content to the existing literature. That is not applied physics at the level we expect. This manuscript appears to contain a presentation of data without the in-depth analysis expected. Hence it is not sufficient to report outcomes of experiments without drawing out new physics in the interpretation of the results. In this manuscript, there are descriptions about the Figures, but there is little said about their physical significance and what new understanding can be gained from them.
Many studies dealing with the absorption properties of composites containing graphene filler particles are not cited and eventually not discussed.
We want to thank the reviewer for his/her comments. In the introduction, we have added and commented some recent papers [17-22] related to microwave absorbing materials based on graphene.
While Figs. 1-4 may be adapted for a technical report, they are irrelevant to be included in a paper of the archival literature.
Figures 1-4 are included in the Materials and Methods Section in order to illustrate to the readers the features of the measured samples, the scheme of the measurement method and the two measurement systems. We think that the figures of this section are useful to clarify to the readers the test procedure used to measure the complex permittivity shown in the results.
Figs. 4-8 are just described and not discussed in physical terms/ What is (are) the physical mechanism(s) of loss? Unclear. What is the physical reason of the wavy nature of the curves in Figs. 6 and 8? Not discussed.
We thank the reviewer for this consideration. We have added the explanation of the physical mechanisms of loss in Section 3 and provided the explanation of the wavy nature of the curve in Fig. 6 and 8.
I do not get the point for the use of Fig. 9. How Eq. (1) is justified in physical terms? Unclear What value of N is chosen and argumented? How many constants (values?) are used in Eq.(1)?
As for Eq (1), we have changed the equation of the general model introduced in the previous version with the model actually applied for our material, well represented by the first order Debye model. We have specified all the parameters of the equation and added a reference [47] where the model is described in its details.
Reviewer 2 Report
The work was properly designed and the overall merits of this manuscript is high. However, I have two concerns:
- The first paragraph of the manuscript is a little obvious, and, in my opinion, it does not help to capture the attention of the readers. I think this paragrapht can be improved.
- An EM simulation is provided in The 3.3 section, but statistical paremeters were not informed. The authors should inform in supplementary information more details related to the simulation.
Author Response
The work was properly designed and the overall merits of this manuscript is high. However, I have two concerns:
- The first paragraph of the manuscript is a little obvious, and, in my opinion, it does not help to capture the attention of the readers. I think this paragrapht can be improved.
We thank the reviewer for this comment. We have extended the first paragraph by adding some recent papers related to the applications of graphene composites in different engineering application fields. In particular, we have emphasized the recent interest in developing microwave absorbing materials based on graphene for high performance electromagnetic wave attenuation.
- An EM simulation is provided in The 3.3 section, but statistical paremeters were not informed. The authors should inform in supplementary information more details related to the simulation.
We have added some useful information in order to clarify the numerical simulations performed. In particular:
- the model of the material has been described through the Debye model [47] and all the parameter values have been reported;
- the solver used in the simulation has been specified and briefly described;
- the boundary conditions for the bounding box have been pointed out;
- the type and the number of the meshcells have been indicated.
Reviewer 3 Report
In this paper, microwave characterization of compounds, based on nylon 6 containing different percentage of graphene nanoplatelets, was carried out taking the neat matrix sample processed under the same conditions as benchmark. The measurements were carried out using two microwave systems, operating at two different frequency bands, appropriate to characterize solid and compact material samples. The achieved results, in line with limited data from the literature and from material data sheets, highlight the possibility to use the present polymers as an excellent electromagnetic interference shielding, as confirmed by the full wave electromagnetic numerical simulations that were conducted with a numerical electromagnetic software.
The topic of the paper is interesting and the researchers in this field of work would have interest to read it. The paper is well structured. However, it still needs some improvements before any final decision.
- The introduction should to be enriched by discussing some recent papers related to the applications of graphene/nanoplatelets composites in industry and different engineering fields. For example, it is suggested to discuss the following articles in the field:
- Zuo, Yu-Ting; Liu, Hong-Jun, Fractal approach to mechanical and electrical properties of graphene/sic composites, Facta Universitatis-Series Mechanical Engineering. 2021, 19(2): 271-284.
- He, J.-H. A new proof of the dual optimization problem and its application to the optimal material distribution of SiC/graphene composite. Reports in Mechanical Engineering 2020, 1(1): 187-191.
- Le NL, Nguyen TP, Vu HN, Nguyen TT, Vu MD, An analytical approach of nonlinear thermo-mechanical buckling of functionally graded graphene-reinforced composite laminated cylindrical shells under compressive axial load surrounded by elastic foundation, Journal of Applied and Computational Mechanics. 2020; 6(2): 357-72.
- The authors mentioned that "The dielectric properties of fabricated samples have been measured using the transmission line method that allows to obtain broadband measurement of the dielectric parameters". I the best interest of readers, I suggest to briefly introduce the " transmission line method ".
- What are the main advantages of the "vector error correction" applied in section 3.1?
- Please discuss the physical meanings of the real and imaginary parts of complex permittivity.
- Please discuss explicitly the assumptions/limitations of the considered model.
- Referring to Fig. 10, I suggest to add further discussions related to this statement "Results show that a brick of thickness t = 4 cm can attenuate the EM field of about 20 decibels around 9 GHz".
Author Response
In this paper, microwave characterization of compounds, based on nylon 6 containing different percentage of graphene nanoplatelets, was carried out taking the neat matrix sample processed under the same conditions as benchmark. The measurements were carried out using two microwave systems, operating at two different frequency bands, appropriate to characterize solid and compact material samples. The achieved results, in line with limited data from the literature and from material data sheets, highlight the possibility to use the present polymers as an excellent electromagnetic interference shielding, as confirmed by the full wave electromagnetic numerical simulations that were conducted with a numerical electromagnetic software.
The topic of the paper is interesting and the researchers in this field of work would have interest to read it. The paper is well structured. However, it still needs some improvements before any final decision.
- The introduction should to be enriched by discussing some recent papers related to the applications of graphene/nanoplatelets composites in industry and different engineering fields. For example, it is suggested to discuss the following articles in the field:
- Zuo, Yu-Ting; Liu, Hong-Jun, Fractal approach to mechanical and electrical properties of graphene/sic composites, Facta Universitatis-Series Mechanical Engineering. 2021, 19(2): 271-284.
- He, J.-H. A new proof of the dual optimization problem and its application to the optimal material distribution of SiC/graphene composite. Reports in Mechanical Engineering 2020, 1(1): 187-191.
- Le NL, Nguyen TP, Vu HN, Nguyen TT, Vu MD, An analytical approach of nonlinear thermo-mechanical buckling of functionally graded graphene-reinforced composite laminated cylindrical shells under compressive axial load surrounded by elastic foundation, Journal of Applied and Computational Mechanics. 2020; 6(2): 357-72.
We are grateful to the reviewer for his comments and suggestions. We have extended the introduction adding and commenting the articles [14-16] recommended.
- The authors mentioned that "The dielectric properties of fabricated samples have been measured using the transmission line method that allows to obtain broadband measurement of the dielectric parameters". I the best interest of readers, I suggest to briefly introduce the " transmission line method ".
The transmission line method has been introduced and discussed giving more details in Section 2.1. Moreover, for further information reference [28], that is the NIST Technical Note where the method is extensively analyzed, has been added.
- What are the main advantages of the "vector error correction" applied in section 3.1?
The main advantages of vector error correction have been inserted in Section 2.2 related to the Experimental Set-Up.
- Please discuss the physical meanings of the real and imaginary parts of complex permittivity.
We have added a phrase on the physical meanings of ε' and ε'' in Section 2.1.
- Please discuss explicitly the assumptions/limitations of the considered model.
The assumptions of the considered transmission line permittivity measurement method are discussed at the beginning of Section 2.1.
- Referring to Fig. 10, I suggest to add further discussions related to this statement "Results show that a brick of thickness t = 4 cm can attenuate the EM field of about 20 decibels around 9 GHz".
We have added a comment to figure 10 related to the shielding effectiveness (SE) of the used composite material; SE is typically defined as the ratio of the magnitude of the incident electric field, to the magnitude of the transmitted electric field. A review reference on Electromagnetic Shielding Effectiveness of Composite Materials has been added [48].
Reviewer 4 Report
This paper reports on a batch-producible electromagnetic shielding composite material and details a measurement method for measuring electromagnetic shielding. The paper has been written well. However, certain concerns need to be noticed and clarified:
- More methods of preparing electromagnetic shielding composites should be cited and discussed in introduction. More related references about functional polymer composites also should be cited for better understanding.
- What are the advantages of this electromagnetic shielding composite material you have prepared compared to theirs in terms of electromagnetic shielding. Need to add a comparison with EM shielding composites prepared by others.
- What are the mechanical properties of this electromagnetic shielding composite? Can it meet the requirements of use?
- What are the electrical conductivity properties of this composite material you have prepared? You need to add the conductivity data of the composite.
- There are some grammar and word mistakes in the manuscript. Please go through the manuscript carefully again.
Author Response
1) More methods of preparing electromagnetic shielding composites should be cited and discussed in introduction. More related references about functional polymer composites also should be cited for better understanding.
Methods for preparing polymer-based EMI shielding materials are presented and discussed in the introduction. Moreover, we have added some recent papers related to the presented methods in the References Section:
- Wang, L, Ma, Z, Zhang, Y, Chen, L, Cao, D, Gu, J Polymer-based EMI shielding composites with 3D conductive networks: A mini-review. SusMat. 2021; 1: 413– 431
- Wenwen Guo, Yuyu Zhao, Xin Wang, Wei Cai, Junling Wang, Lei Song, Yuan Hu, Multifunctional epoxy composites with highly flame retardant and effective electromagnetic interference shielding performances, Composites Part B: Engineering, Volume 192, 2020, 107990, ISSN 1359-8368,
- Yue-Yi Wang, Zi-Han Zhou, Chang-Ge Zhou, Wen-Jin Sun, Jie-Feng Gao, Kun Dai, Ding-Xiang Yan, and Zhong-Ming Li, Lightweight and Robust Carbon Nanotube/Polyimide Foam for Efficient and Heat-Resistant Electromagnetic Interference Shielding and Microwave Absorption, ACS Applied Materials & Interfaces 2020 12 (7), 8704-8712 DOI: 10.1021/acsami.9b21048.
- Zhao S, Yan Y, Gao A, Zhao S, Cui J, Zhang G. Flexible polydimethylsilane nanocomposites enhanced with a threedimensional graphene/carbon nanotube bicontinuous framework for high-performance electromagnetic interference shielding. ACS Appl Mater Interfaces. 2018; 10:26723-26732.
- Kuang T, Chang L, Chen F, Sheng Y, Fu D, Peng X. Facile preparation of lightweight high-strength biodegradable poly-mer/multi-walled carbon nanotubes nanocomposite foams for electromagnetic interference shielding. Carbon. 2016; 105:305-313.
- Zhaoxin Xie, Yifan Cai, Yanhu Zhan, Yanyan Meng, Yuchao Li, Qian Xie, Hesheng Xia. Thermal insulating rubber foams embedded with segregated carbon nanotube networks for electromagnetic shielding applications. Chemical Engineering Journal 2022, 435 , 135118.
- Zhiqiang Su, Li, JS., Huang, H., Zhou, YJ. et al. Research progress of graphene-based microwave absorbing materials in the last decade. Journal of Materials Research 32, 1213–1230 (2017). https://doi.org/10.1557/jmr.2017.80.
2) What are the advantages of this electromagnetic shielding composite material you have prepared compared to theirs in terms of electromagnetic shielding. Need to add a comparison with EM shielding composites prepared by others.
The proposed composite material has a shielding effectiveness value comparable and, in some cases, higher than shielding effectiveness of composite materials reported in reference [55], where an overview of the SE of different composites materials is given. We have added some comments on the SE of other papers in Section 3.3.
Moreover, we have added some information on the mechanical performances of considered composites that have been studied in [59], showing that the combination of the two materials proposed in the paper leads to an enhanced material in terms of electrical and mechanical properties, following actual trends of economical and environmental global strategies.
3) What are the mechanical properties of this electromagnetic shielding composite? Can it meet the requirements of use?
The mechanical properties of the composites considered were the subject of a previous paper (P. Russo, F. Cimino, A. Tufano, F. Fabbrocino Thermal and quasi-static mechanical characterization of polyamide 6-graphene nanoplatelets composites Nanomaterials 11, 1454 (2021)).
In particular, Russo et al. showed that the inclusion of GNPs, on the range explored so far, generates a monotonous growth of the host matrix stiffness both in tensile (+ 13%) and, above all, in bending with average values increasing from 1196 MPa for neat polyamide to 3303 MPa (+ 276%) for the sample filled with 5 wt% -of GNPs.
Vice versa, for situations of quasi-static loading, traction and bending, a non-monotonous trend of the strength was detected with average values maximized for the sample containing 3% by weight of GNP and in any case, in bending, always higher than the value characterizing the neat matrix.
Considering the already consolidated wide use of polyamides 6 in many industrial sectors such as transport and electronics in which the employment of EMI shielding systems is increasingly required, the further improvement of the mechanical performance mentioned above allows us to consider that the new formulations can certainly satisfy the requirements of the reference markets.
4) What are the electrical conductivity properties of this composite material you have prepared? You need to add the conductivity data of the composite.
We have added electrical conductivity properties in Section 3.3 together with some comments on the obtained value.
5) There are some grammar and word mistakes in the manuscript. Please go through the manuscript carefully again.
We tried to correct the errors by carefully reviewing the manuscript.
Round 2
Reviewer 1 Report
The main defects pointed out in my earlier report have not been taken into account. This manuscript is outside the topics of the SI Mathematical and Computational Modelling in Mechanics of Materials and Structures.
This manuscript appears to contain a presentation of data without the in-depth analysis expected. Hence it is not sufficient to report outcomes of experiments without drawing out new physics in the interpretation of the results. In this manuscript, there are descriptions about the Figures, but there is little said about their physical significance and what new understanding can be gained from them.
I cannot recommend this manuscript for publication anywhere.
Author Response
The authors are sorry that despite the revisions made in the first round, the reviewer considers that the manuscript is not publishable.
Reviewer 2 Report
Dear Authors, this reviewer considers that al comment has been replied, and considers that the manuscript quality was further increased.
Best regards
Author Response
We want to thank the reviewer for his/her comments.
Reviewer 3 Report
The authors have suitably addressed all the comments. The paper is recommended for publishing as it is.
Author Response

(The authors gave the same response as above.)

Reviewer 4 Report
Can be accepted now.